# Studying the Plastic Deformation of Cu-Ti-C-B Composites in a Favorable Stress State

**DOI:** 10.3390/ma16083204

**Published:** 2023-04-18

**Authors:** Nataliya Pugacheva, Denis Kryuchkov, Tatiana Bykova, Dmitry Vichuzhanin

**Affiliations:** Institute of Engineering Science, Ural Branch of the Russian Academy of Sciences, 34 Komsomolskaya St., Ekaterinburg 620049, Russia; kru4koff@bk.ru (D.K.); tatiana_8801@mail.ru (T.B.); mmm@imach.uran.ru (D.V.)

**Keywords:** composites, self-propagating high-temperature synthesis, copper matrix, structure, hardness, hot deformation, favorable stress state, dynamic recrystallization

## Abstract

Composites with a copper matrix attract the attention of researchers due to their ability to combine high ductility, heat conductivity, and electrical conductivity of the matrix with the high hardness and strength of the reinforcing phases. In this paper, we present the results of studying the effect of thermal deformation processing of a Сu-Ti-C-B composite produced by self-propagating high-temperature synthesis (SHS) on its ability to deform plastically without failure. The composite consists of a copper matrix and reinforced particles of titanium carbide TiC (sized up to 1.0 μm) and titanium diboride TiB_2_ (sized up to 3.0 μm). The composite hardness is 60 HRC. Under uniaxial compression, the composite starts to deform plastically at a temperature of 700 °C and a pressure of 100 MPa. Temperatures ranging between 765 and 800 °C and an initial pressure of 150 MPa prove to be the most effective condition for composite deformation. These conditions enabled a true strain of 0.36 to be obtained without composite failure. Under higher strain, surface cracks appeared on the specimen surface. The EBSD analysis shows that dynamic recrystallization prevails at a deformation temperature of at least 765 °C; therefore, the composite can plastically deform. To increase the deformability of the composite, it is proposed to perform deformation under conditions of a favorable stress state. Based on the results of numerical modeling by the finite element method, the critical diameter of the steel shell is determined, which is sufficient for deformation of the composite with the most uniform distribution of the stress coefficient k. Composite deformation in a steel shell under a pressure of 150 MPa, at 800 °C, is experimentally implemented until a true strain of 0.53 is reached.

## 1. Introduction

Creation of new composite materials by applying the self-propagating high-temperature synthesis (SHS) technique has been successfully developed for more than 50 years [1,2,3,4,5]. The production of powders for sintering or spraying of coatings remains the guideline in applying the SHS technique [6,7]. SHS has been successfully used to produce gradient materials as heat protection systems for high-temperature applications [8,9]. Creation of monolithic composites that are finished parts or structural components is of interest. Monolithic SHS composites are wear-resistant materials; therefore, the emphasis is generally on strengthening phases, namely carbides and borides, which provide high values of hardness and wear resistance [10,11,12,13,14]. It is still difficult to produce void-free monolithic composites. Using copper as a matrix is of interest both in terms of lower porosity due to hot pressing immediately after the completion of SHS [15,16,17] and in terms of the feasibility of subsequent plastic deformation.

Copper possesses high ductility, heat conductivity, and electrical conductivity [18]. The fairly low melting temperature of copper (1083 °C) [19] minimizes the difference between the actual and theoretical composite densities. Monolithic Cu-Ti-C-B SHS composites have yet to be studied more extensively. The features of the formation of SHS composites with a copper or copper-alloy matrix were studied in [20,21,22]. These composites attracted interest due to the ability to produce new materials with high thermophysical properties and electric conductivity [23,24,25]. However, there are no studies on the plastic deformability of SHS composites. Studying this issue is of practical importance in terms of making products with specified shape and dimensions. It was reported in [26,27,28] that severe plastic deformation of Cu/Mg composites, when combined with heat treatment, allowed unique properties to be obtained due to a combination of high electric conductivity and strength. In [29,30], it was reported that phase transformations in the metal matrix of composites under heating increased ductility during hot deformation. As is known from [31,32], hardening competes with recrystallization during hot plastic deformation of alloys. By defining the conditions for the predominance of recrystallization over deformation processes in Cu-Ti-C-B composites, it is possible to select the conditions of their plastic deformation without failure. Previously published works noted the high hardness and wear resistance of monolithic SHS composites [33,34]. Conditions and mechanisms of plastic deformation of monolithic SHS composites Cu-Ti-C-B have never been studied before. 

In this context, the aim of this study is to determine the temperature and load of stable plastic deformation of a Cu-Ti-C-B composite under uniaxial favorable loading and to perform an EBSD analysis of changes in the composite structure during plastic deformation.

## 2. Materials and Methods

The specimens studied were cut out from a monolithic Cu-Ti-C-B SHS composite produced by the procedure described in [33,34]. Copper metal powders were used to form the composite matrix, of which the particles are shown in Figure 1a. The exothermic reactions were provided by heat-sensitive constituents (HSC), namely powders of titanium (Figure 1b), carbon (Figure 1c), and boron carbide B_4_C (Figure 1d).

The quantitative HSC ratio must ensure the complete course of the following exothermic reactions:Ti + C = TiC + Q,(1)
3Ti + В_4_С = TiC + 2TiB_2_ + Q.(2)

Initial powders were put into a ball mill with a 5 L capacity together with grinding balls made out of steel ShH15. The powder mass ratio in relation to the mass of balls was 1:3. The mixing type was dry and the mixing time was 12 h. A steel pipe container made from a low-carbon structural steel (brand St3, Seversky Pipe Plant, Polevskoy, Russia) was filled with the obtained powder mix. The initial compacting of the powder mix was achieved with a special snap. Then, the initial sample was immersed in an electric oven and heated up to the temperature of an exothermic reaction (1020 °С). After SHS completion, in order to eliminate porosity, the hot sample was transferred to a hydraulic press and compacted with a weight of not less than 250 MPa (Figure 2a). This resulted in sandwich plates with a steel outer shell and a Cu-Ti-C-B composite inside (Figure 2b). The composite matrix is Cu-based solid solution, and the reinforcing particles are titanium carbide TiC and titanium diboride TiB_2_. Some properties of the phases are shown in Table 1. Titanium carbide and titanium diboride are characterized by high strength and resistance to high temperature and corrosive environments. 

A Tescan Vega II XMU scanning electron microscope (Tescan, Brno, Czech Republic) was used to examine the composite structure on the cross-sectional cuts of the sandwich plates (Figure 2c). The local chemical composition of the phases and structural constituents of the composites were determined by means of an Oxford energy dispersive attachment with the INCA software (Oxford Instruments, Abingdon upon Thames, UK). The EBSD analysis was performed using the software for an Oxford HKL Nordlys F+ accessory. The phase X-ray diffraction analysis was performed in a Shimadzu XRD-700 diffractometer (Shimadzu Scientific, Kyoto, Japan) in monochromatized Kα radiation of a chromium anode in the diffraction angle range of 20°–85°. Rockwell hardness was measured with a hardness tester on the HRC scale by ISO 6508-86 [37]. The values of the thermal linear expansion coefficient (TLEC) of the Сu-Ti-C-B composite were determined experimentally using a Netzsch DIL 402C dilatometer (Netzsch, Selb, Germany) with the use of a highly sensitive linear displacement transducer under heating at a constant rate of 4 °C/min in highly pure helium.

The micromechanical properties of the composite were determined according to GOST R 8.748-2011 (ISO 14577-1:2002) [38]. Instrumented microindentation with recording of the loading diagram was performed by means of a Fischerscope HM 2000 XYm (Helmut Fischer GmbH + Co. KG, Stuttgart, Germany) device with a Vickers indenter and the WIN-HCU 3.9 software at a maximum load of 0.980 N, a loading time of 20 s, a holding time under load of 15 s, and an unloading time of 20 s. The error in the microhardness and microhardness characteristics over 10 measurements was calculated with a confidence probability *p* = 0.95. The following micromechanical properties of the structural constituents of the composite were determined from the results of indentation: Vickers microhardness (НV), the contact elastic modulus (E*), the work of plastic deformation during indentation (φ), creep during indentation (С_IT_), the elastic recovery factor (R_e_), and the share of elastic strain in the total strain during indentation (H_IT_/E*) [39,40,41], where H_IT_ denotes the values of indentation hardness under maximum load. The quantities φ, С_IT_, and R_e_ were calculated by the following formulas:φ = (1 − W_e_/W_t_) × 100%, (3)
С_IT_ = (h_max_ − h_1_)/h_1_ × 100%, (4)
R_e_ = [(h_max_ − h_p_)/h_max_] × 100%, (5)
where W_e_ is the work of elastic deformation during indentation which is released at unloading; W_t_ is the total mechanical work during indentation, which is determined by the area under the loading curve; h_1_ is the indentation depth corresponding to the initial point of the horizontal length of the loading curve; h_p_ is the depth of impression after unloading; h_max_ is the maximum indentation depth. Determination of the micromechanical properties is necessary to identify the most plastic microconstituent that can provide plastic deformation of the composite as a whole.

Uniaxial deformation of the composite was performed on rectangular parallelepiped specimens, with a base side (a) of 5 mm and a height (h_0_) of 8 mm, cut out from the central part of the sandwich plates. A graphite lubricant was applied onto the specimen bases (ends) contacting surfaces with the tool. During the test, each specimen was placed in a cold furnace and heated up to a specified temperature (Figure 3a). The heating temperature was monitored by a chromel-alumel thermocouple mounted on the specimen. During the heating, the specimen was affected by pressure specified by the force on the plunger of the Instron 8801 testing machine (Figure 3b). During the deformation, the pressure decreased due to the increasing contact area; therefore, the process was nonstationary. The experiments were made under initial pressures ranging between 100 and 150 MPa with heating to 700–900 °C. The heating schedules are shown in Figure 4.

The plunger displacement, the time, and the specimen temperature were monitored during the tests. The change in the specimen height was determined by punch displacement during the testing. The specimen dimensions before and after testing were measured using a micrometer and a toolmaker’s microscope, and then, true strain (е) and process average strain rate (ξ*) were calculated using the following formulas:e = ln(h_0_/h_1_),(6)
ξ* = e/t.(7)

A pressure of 10 MPa was applied to the specimen heated to 700 °С, and then the readings of the punch displacement sensor were taken. By gradually increasing the load, the pressure on the specimen was changed with a step of 5 MPa until the start of punch displacement was recorded. This procedure enables one to determine the conditions under which a material starts to deform.

## 3. Results

### 3.1. Microstructure, X-ray, and EBSD Analysis 

The Cu-Ti-C-B composite under study consists of a metal matrix (a copper-based solid solution) and a small quantity of titanium (about 2 at.% according to a local X-ray microdiffraction analysis) dissolved in its crystal lattice. Grey globular particles of titanium carbide (TiC, sized from 0.1 to 1.0 μm) and black cubic particles of titanium diboride (TiB_2_, sized from 0.5 to 3.0 μm), resulting from the synthesis, are the strengthening phases (Figure 5a). The TiC particles in a quantity of about 14 wt% are uniformly distributed in the composite volume, and the TiB_2_ particles are distributed nonuniformly; the TiC particles form clusters, and their quantity is about half that of the TiC particles.

The phase X-ray diffraction analysis revealed four phases in the Cu-Ti-C-B composite (Figure 5b). In addition to TiC and TiB_2_, weak reflections of the Cu_4_Ti intermetallic compound were recorded. Homogeneous precipitation of Cu_4_Ti nanoparticles in a Cu-based solid solution supersaturated with Ti has been studied earlier and discussed in [42]. Therefore, in the following discussion related to a copper matrix, we mean a Cu-based solid solution containing Cu_4_Ti nanoparticles. The copper matrix acts as a binder in the composite. This is the most plastic component, and therefore, it can withstand external mechanical loads. The Cu-based solid solution is capable of relieving the stresses that occur during the manufacturing of the sandwich plate. Moreover, it is the copper matrix that provides the electrical and thermal conductivity of the composite. Nanosized particles harden the matrix but do not reduce its plasticity. The small size of Cu_4_Ti particles (10 nm [42]) does not prevent the motion of dislocations in the copper crystal lattice. Dislocations cut these particles easily. Reinforcing TiC and TiB_2_ particles increases the hardness and strength of the composite, thereby increasing wear resistance [43]. TiB_2_ particles will somewhat reduce the electrical and thermal conductivity of the copper matrix (Table 1); however, TiC particles should not affect these physical characteristics. The positive effect of reinforcing particles is to improve the thermal stability and corrosion resistance of the composite. Determination of electrical and thermal conductivity, corrosion resistance, and thermal stability requires additional studies.

It was rather difficult to separate each of these phases during the EDS analysis. The results presented in Table 2 determine the chemical composition of three structural constituents of the composite. Each constituent is a mechanical mixture of two or three phases. The mechanical mixture of the Cu-based solid solution and the titanium carbide particles, i.e., Cu + TiC, occupies the main volume of the composite (marked by “1” in Figure 5a). It is known from [36] that titanium carbide TiC is isomorphic to titanium boride TiB, and these chemical compositions are absolutely mutually soluble. In this connection, the TiC particles contain some quantity of boron, and this is reflected in the chemical composition of Constituent 1 (Table 2). Near the TiB_2_ particles, most often, there are smaller TiC particles; therefore, the second structural constituent was revealed, ”Cu + TiB_2_ + TiC” (marked by “2” in Figure 5а). The third structural constituent is a ”Cu + TiB_2_“ mechanical mixture, containing practically no TiC particles (marked by “3” in Figure 5a).

The basis of the copper matrix of the Cu-Ti-C-B composite consists of grains that have a subcrystalline structure (68%) with a preponderance of low-angle boundaries (Figure 6a–c). The share of deformed grains is 26%, and the share of recrystallized grains is 6% (Figure 6d). This testifies that dynamic recrystallization occurs during hot pressing of the composite. In order to estimate the Cu-Ti-C-B composite texture, the coordinate system in the EBSD analysis was selected as follows: the Х axis is aligned with the deformation direction during composite pressing (DD); the Y axis is directed across the sandwich plate (AD); the Z axis is directed along the sandwich plate, normal to the direction of deformation (ND). A combined crystallographic texture was revealed for the metal matrix of the Cu-Ti-C-B composite. Along the Х axis, it is the <112> directions that prevail in the texture, and the texture of the <102> directions is less pronounced (Figure 6b). Along the Y direction, we observe a diffuse texture, the predominance of the <235> direction, and a lower density of the direction close to <302>. The <223> orientation prevails along the Z axis, whereas the other orientations <331> and <731> have a much lower pole density.

### 3.2. Micromechanical Properties

Micromechanical properties were determined for the three constituents singled out in the composite and are marked in Figure 5a. The Cu + TiC regions are characterized by the lowest microhardness ranging between 400 and 510 HV 0.1; the average value is 450 HV 0.1. The microhardness of the Cu + TiB_2_ + TiC regions ranges between 625 and 800 HV 0.1; the average value is 760 HV 0.1. The Cu + TiB_2_ regions are characterized by the highest microhardness ranging from 900 to 1150 HV 0.1; the average value is 1050 HV 0.1 (Table 3). The integral hardness of the composite is 60 HRC.

The nonuniform distribution of the strengthening phases in the composite volume is responsible for the nonuniform distribution of the micromechanical properties. The loading diagrams for the three structural constituents of the Cu-Ti-C-B composite are separated and positioned in the regions characterized by the largest indentation depth h_max_, which, in turn, determines the total deformation of the composite under indentation. The Cu + TiC regions are the most plastic ones, and their corresponding loading diagrams, shown in Figure 7, are shifted to the right-most position. This structural constituent is characterized by the highest values of plasticity indices h_max_, φ, and C_IT_ (Table 3). The highest strengthening is observed in zones with the predominance of TiB_2_ particles, such as Area 3 in Figure 5а. The corresponding loading diagram is shifted to the left-most position (Figure 7). The highest values of the hardening parameters H_IT_, HV, W_е_, R_e_, and H_IT_/E* correspond to this structural constituent. 

The contact elastic modulus E* in the composite regions enriched with strengthening phase particles increases significantly due to the predominance of the phases with a large contribution of the covalent component in the lattice. The H_IT_/E* ratio characterizes the contribution of elastic strain in total strain under indentation, and it indirectly characterizes the wear resistance of the composite constituents [41]. As one would expect, the high content of TiB_2_ particles in the composite has the most noticeable effect on an increase in this ratio (Table 4). The Cu + TiC + TiB_2_ constituent, marked by “2” in Figure 5a, is intermediate in terms of the micromechanical properties (Figure 7 and Table 3). 

### 3.3. Uniaxial Deformation under Nonstationary Conditions

The minimum threshold pressure at which the deformation of the Cu-Ti-C-B composite starts is 100 MPa. A 25 MPa increase in the initial pressure results in deformation at 35–60 °C lower heating temperature (Table 4). At an initial pressure of 100 MPa, the value 0.6 of true strain can be achieved when the specimen is heated to a temperature exceeding 875 °C. By increasing the pressure to 125 MPa, it is possible to lower the specimen heating temperature to 825 °C and to obtain a strain of 0.6. An increase in the initial pressure on the specimen to 150 MPa enables us to deform the composite even at 700 °C and to obtain a true strain exceeding 0.3 at 765 °C. According to the temperature dependence of strain (Figure 8a), the higher the initial pressure, the more rapid the accumulation of strain. When the initial pressure on the specimen is 100 MPa, essential strain accumulation from 0.15 to 0.76 is smooth within the time of heating from 850 to 900 °C. When the initial pressure is 150 MPa, strain accumulation from 0.4 to 0.83 is stepwise at 800 °C.

On the surfaces of the specimens deformed under Conditions 1, 2, 6, and 7, there are no cracks, and there are traces of plastic flow in the form of deformation bands (Figure 9a). The highest strain (e = 0.36) without fracture is achieved on Specimen 7 at a heating temperature of 765 °C under an initial pressure of 150 MPa (Figure 8a). On the side surface, there are single micro-delaminations near the TiB_2_ particle clusters (Figure 9b). The peak strain rate values for the specimens with the strain e > 0.6 appeared at 800 and 870 °C (Figure 8b). In these cases, the strain rate reaches values exceeding 0.002 s^−1^. These jumps are caused by cracking on the surfaces of the specimens (Figure 9c). For the composite to deform without cracking, the strain rate must not exceed 0.0005 s^−1^. Herewith, the process average strain rate must not exceed 7.5∙10^−5^ s^−1^.

The ЕBSD analysis has shown that, at a deformation temperature of 700 °C and a load of 150 MPa, a deformed structure prevails in the composite and recrystallization develops mainly in the initial stage, i.e., the stage of polygonization. The subcrystalline grains of the matrix occupy 31% of the total volume, and the share of fully recrystallized grains is only 6% (Figure 10). 

Increasing the deformation temperature to 765 °C causes an increase in the share of recrystallized grains. At that, during deformation under a lower pressure of 100 MPa, dynamic polygonization prevails, namely, the majority of the copper matrix grains have low-angle boundaries (Figure 11a–c), grains with a subcrystalline structure form the major portion (92%), and the share of deformed grains amounts to only 3%. (Figure 11d). A higher load of 150 MPa increases the number of grains with high-angle boundaries (Figure 11e,f), doubles the share of recrystallized grains, and decreases considerably the number of subgrains; the main volume of the composite is occupied by deformed grains (Figure 11g). Note that, at temperatures of at least 765 °C, oxidation processes develop on the specimen surface.

Thus, even when the Cu-Ti-C-B SHS composite is deformed hot, one faces the problem of surface cracking. The maximum logarithmic strain without fracture that we managed to obtain during the compression testing of the composite was 0.36.

### 3.4. Deformation in a Favorable Stress State

To be able to deform a composite without fracture, it is necessary to create a favorable stress state, for example, to deform under conditions of high hydrostatic pressure [44]. There have been studies on producing composites by SHS extrusion [45]. This technique involves hydrostatic compression of a composite during synthesis and subsequent complex shearing of the composite material immediately after synthesis completion, which combines the translational and rotational motions.

The stress coefficient (or triaxiality factor) is currently widely used to evaluate hydrostatic stress [46,47,48,49,50,51] as follows:*k* = σ/*T*,(8)
where σ is the mean normal stress and *T* is the tangential stress intensity. The mean normal stress is evaluated by the formula:σ = 1/3(σ_11_ + σ_22_ + σ_33_), (9)
where σ_11_, σ_22_, and σ_33_ are principal stresses. Tangential stress intensity is defined as: (10)$$T=\sqrt{0.5S_{ij}S_{ij}},\;$$
where *S_ij_* denotes stress deviator components. Tensile stresses prevail when *k* is positive, while compressive stresses are predominant when *k* is negative. The lower the value of *k*, the more favorable is the stress state in the material under study and the higher is the strain to failure it can sustain.

Thus, if the composite is deformed in a favorable stress state, it sustains considerable plastic strains without failure. In this connection, a specimen of the composite was tested for compression in a steel shell (Figure 12a). The specimen (1) sized 5 × 5 × 6 mm was placed into a shell (2) made from medium carbon steel of grade 40. The whole structure was then mounted onto the bottom die (4) of the Instron 8801 testing machine. The top die (3) translates the force from the testing machine to the specimen. The feature of this type of testing is that, since the specimen is tested at a high temperature, the specimen and the shell have significantly different thermal linear expansion coefficients (TLEC); therefore, before testing at 800 °C, the specimen is under conditions of a favorable stress state. The TLEC values for steel 40 are selected from the steel grade guide [52] (p. 589), and they were found experimentally for the Сu-Ti-C-B composite (Table 5). 

In order to evaluate the stress-strain state of the composite, its deformation (compression in a steel shell) was simulated by using the finite element method with the ANSYS Academic Research v. 16.2 software on the URAN cluster computer. A 3D finite element model was constructed; since the specimen and the shell were symmetric, only their quarters were simulated (Figure 12b). An isotropic elastic–plastic model with strain hardening was assumed for the material of the specimen to be deformed. The material of the dies was assumed to be a perfectly rigid body. The associated plastic flow rule and the von Mises yield condition were applied. The boundary conditions were set in displacements. On two symmetry planes, the following constraints of specimen and shell displacements in the corresponding directions are given: *u_x_* = 0 and *u_z_* = 0 (Figure 12b). The lower die is fixed from moving in three directions: *u_x_* = 0, *u_y_* = 0, and *u_z_* = 0. The upper die is fixed from moving in two directions: *u_x_* = 0 and *u_z_* = 0. In the direction of load application, displacement is given as *u_y_* = 3 mm (Figure 12b). The friction between the deforming tool and the specimen in the shell was described by the Amontons–Coulomb friction law. The friction coefficient was taken to be 0.1. 

The simulation results enable us to estimate the distribution of the values of the stress coefficient (*k*) in the composite. The values of the coefficient are low, and this means that compressive stresses dominate in the composite during the testing. However, note that, in the simulation of compression in a shell with a diameter of 10 mm, the distribution of the values is rather nonuniform (Figure 12c). As the shell diameter increases, the distribution of the stress coefficient values becomes more uniform and the stress state becomes more favorable (Figure 12d). Table 6 shows the volume-averaged values of the stress coefficient *k*_av_, obtained from numerical simulation, as dependent on the shell diameter. As the shell diameter increases, the decrease of *k*_av_ slows down. The use of a steel shell with a diameter exceeding 9 mm leaves the parameter *k*_av_, characterizing the nonuniformity of the distribution of *k* in the composite volume, practically unchanged. In view of the data from Table 6, a 10 mm diameter shell was used for the experiment.

Figure 13a shows a photograph of the Cu-Ti-C-B composite specimen in a shell. The deformation was performed in an Instron 8801 servohydraulic testing device supplemented with a heating furnace. The specimen was put into a cold furnace and heated to 800 °C after application of a load of 150 MPa. The displacement of the traverse of the testing machine, the time, and the specimen temperature were recorded during heating. The testing was completed after the heating temperature had been achieved. The specimen was heated to the required temperature of 800 °C within 80 min (Figure 13b). The deformation starts at 600 °С (Figure 13c). The temperature dependence of strain for the composite has four portions differing in strain rate. Between 710 and 750 °C (2 in Figure 13c), the strain rate approximately doubles from that in the initial portion marked by “1”. In Portion 3, at a temperature of about 750 °C, there is a small interval of about 15 °C where the specimen is not deformed. At higher temperatures in Portion 4, the strain rate is the same as in Portion 2. The total strain is 0.53.

The EBSD analysis of the composite after hot deformation in a shell has shown that deformed grains dominate in the copper matrix (Figure 14a), amounting to 43% on average. However, the subcrystalline structure with low-angle boundaries (Figure 14b,c), resulting from the stage of polygonization (recrystallization stage I), occupies a slightly smaller volume of 39%. The share of fully recrystallized grains is fairly significant, namely 18% (Figure 14d). The hot deformation of the composite under conditions of uniform compression results in the predominant orientation of the <116> axes along the direction of applied load (the X-axis) and the predominant orientation of the <104> crystallographic axes along the Y-axis, i.e., across the sandwich plates; no predominant orientation of crystallographic axes was noticed along the sandwich plates (the Z-axis) (Figure 14e). 

## 4. Discussion

The Cu-Ti-С-B composite is characterized by a fairly high content of the strengthening phases of titanium carbide and titanium diboride particles, namely at least 21 wt%. The copper matrix is a supersaturated solid solution of titanium in a copper lattice; this ensures its solid-solution hardening. Nanosized Cu_4_Ti particles, whose morphology was studied in [42], precipitated in the supersaturated copper-based solid solution. After the synthesis had been completed, the composite was hot pressed; therefore, naturally, the copper matrix became strain hardened. All these factors are responsible for the high integral hardness of the composite and, consequently, for high wear resistance indices, as was reported in [43]. Plastic deformation of such hardened composites is difficult to perform. It was demonstrated in [27,29,30,31] that hardened composites could be plastically deformed only at high temperatures. 

Instrumented microindentation testifies that the Cu + TiC constituent manifests the ability to deform plastically (Table 3 and Figure 7). This is due to the fact that the lattice of titanium carbide particles is closer than that of titanium diboride particles to the lattice of the metal matrix. Titanium carbide TiC, as well as the copper-based solid solution, has a B1 (NaCL-type) FCC lattice belonging to the Fm3m space group, whereas titanium diboride TiB_2_ has a С32 (AlB2-type) hexagonal lattice belonging to the Р6/mmm space group, which explains why the contact elastic modulus E* is much higher in the regions containing titanium diboride particles than in the Cu+TiC regions, this being due to a larger effect of the covalent component of the TiB_2_ lattice, which ensures a more rigid interatomic bond. 

The EBSD analysis reveals the first stage of recrystallization, i.e., polygonization in the copper matrix of the initial composite with a predominance of low-angle boundaries due to the arrangement of dislocations into walls. The fraction of deformed grains in the copper matrix is 26%, the main volume (68%) consists of grains with a subcrystalline (polygonized) structure, 6% being occupied by fully recrystallized grains. As is known from [31,32], two processes compete in the hot deformation of metallic materials, namely strain hardening and recrystallization-induced softening. In the composite under study, recrystallization can occur both during hot pressing and during subsequent cooling since, some time after load removal, the composite retains a temperature above 0.5 T_melt_, sufficient for recrystallization to develop. Thus, for the Cu-Ti-C-B composite, plastic deformation is feasible due to the presence of grains with minimum dislocation density in the copper matrix. The diffuse texture of the composite matrix (Figure 6d) is not an obstacle for the motion of dislocations in the polygonized and recrystallized grains. 

This research testifies that the processes occurring in the Cu-Ti-C-B composite during thermal deformation processing are much more complex. On the one hand, heating to about 700–800 °С and higher causes recrystallization and softening in the composite, but on the other hand, diffuse processes become more active, and this may result in partial dissolution of the strengthening phases. Herewith, the contribution of precipitation hardening decreases, but there occurs solid solution hardening. Hardening and softening compete depending on the temperature and the amount of composite strain. Thus, at 700 °С, under a pressure of 150 MPa, deformation processes obviously prevail (Figure 10), i.e., the fraction of deformed grains in the matrix is 63%, the fraction of grains with a subcrystalline structure is 31%, the recrystallized structure occupies 6%, and the true strain of the specimen amounts to 0.13. It is evident that the diffusion processes in the composite are insufficiently developed at 700 °C; therefore, the alloying level of the solid solution remains unchanged. At this temperature, the recrystallization processes are mainly in the initial stage, that of polygonization. At a higher temperature of 765 °C, an increase in the diffusive mobility of the chemical atoms in the composite should be expected, as well as an increase in the dislocation velocity. Under a pressure of 100 MPa, recrystallization processes dominate in the metal matrix since the main volume of the composite is occupied by grains with a subcrystalline structure (92%). The fraction of deformed grains is as small as 3% (Figure 11a–d). The pressure increase to 150 MPa at the same temperature activates deformation processes (Figure 11e–g).

A further increase in the temperature to 800 °C makes diffusion processes even more active and increases the probability of partial dissolution of the strengthening phases. The alloying elements go into the copper-based solid solution, and the degree of hardening increases. Thus, several processes compete: The hardening of the solid solution increases due to its higher alloying, whereas precipitation hardening decreases due to the smaller sizes of the dispersed strengthening phases or their complete dissolution. After deformation at a temperature of at least 765 °C, recrystallized structures prevail in the composite matrix under both nonstationary uniaxial compression and favorable loading conditions (Figure 14). Thus, compression at a temperature of at least 765 °C creates conditions for plastic deformation of the composite.

Under nonstationary uniaxial compression, the stresses in the composite are distributed nonuniformly and the high values of compressive stresses on the specimen surface cause cracking. Therefore, under conditions of uniaxial compression, we failed to obtain true strain without failure higher than 0.36. The finite element numerical simulation of the uniform compression of the composite under heating to 800 °C with an initial load of 150 MPa has shown that, for a favorable stress state to be implemented, it would suffice to use a 10 mm thick steel shell. The plastic deformation of the composite in the shell starts after the formation of a sufficient portion of softened grains in the copper matrix, which are fully recrystallized grains and those with a subcrystalline structure. The beginning of plastic deformation was recorded as early as at 600 °C. Evidently, dynamic recovery dominates in Portion 1, which manifests itself in the formation of a subcrystalline structure. At a temperature of at least 600 °C, the TLEC of the composite is essentially higher than that of the steel. Therefore, the composite is under additional compressive load. At temperatures exceeding 700 °C, diffusion processes become more active, the motion and interaction of dislocations in the copper matrix of the composite are easier, recrystallization processes develop, and the plastic deformation of recrystallized grains is faster (Portion 2 in Figure 13c). Deformation deceleration at temperatures between 710 and 750 °C may be due to the effect of phase transformations in the steel shell (portion 3 in Figure 13c). It is in this temperature interval that the pearlite-ferrite structure of the steel turns into austenite. With a further temperature increase (Portion 4 in Figure 13c), the copper matrix undergoes dynamic recrystallization, and the plastic deformation of the composite is resumed at the same rate as in Portion 1. 

## 5. Conclusions

A complex analysis of the structure, micromechanical properties, and texture of a Cu-Ti-C-B SHS composite has shown that, despite a fairly large (at least 21%) number of strengthening phases (TiC, TiB_2_, and Сu_4_Ti) in the copper matrix and a high integral hardness (60 HRC), plastic deformation without failure is possible if certain conditions are met. Firstly, plastic deformation of the composite is ensured by the formation of a sufficient amount of a recrystallized structure with a prevalence of low-angle boundaries of copper matrix grains. This can be achieved when composite deformation is performed under uniaxial compression at a pressure of 150 MPa and a temperature ranging between 765 and 800 °C, with a true strain of 0.36. Secondly, the determining condition for the plastic deformation of the composite without surface cracking is uniform compression in a favorable stress state. The true strain of the composite has been experimentally obtained to be e = 0.53 at a deformation temperature of 800 °C under a pressure of 150 MPa.

## Figures and Tables

**Figure 1 materials-16-03204-f001:**
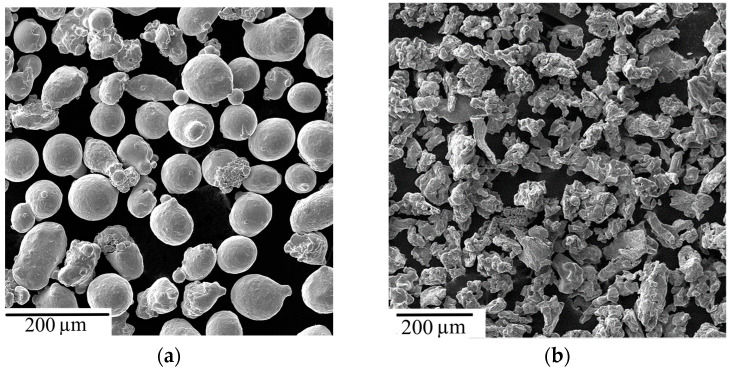
Powders intended for producing a Cu-Ti-C-B SHS composite μm: (**a**) copper; (**b**) titanium; (**c**), carbon; (**d**) boron carbide.

**Figure 2 materials-16-03204-f002:**
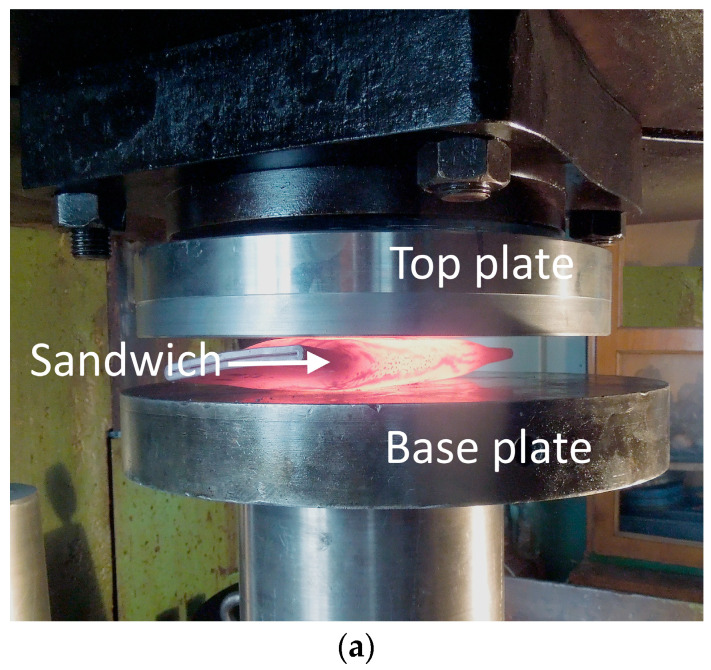
(**a**) Hot pressing of the billet after completion of SHS; (**b**) the exterior of the sandwich plate; (**c**) the specimen for studying the microstructure and the micromechanical properties (1—composite and 2—steel shell).

**Figure 3 materials-16-03204-f003:**
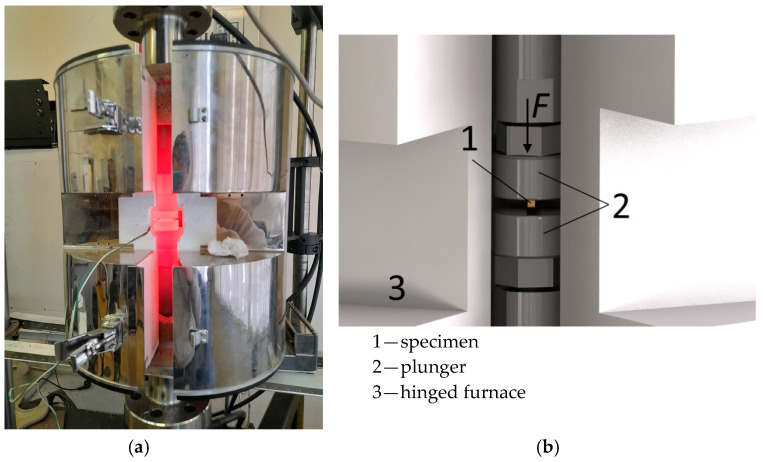
The exterior of the installation (**a**) and the specimen position (**b**) for the uniaxial nonstationary deformation tests of the Cu-Ti-C-B specimen.

**Figure 4 materials-16-03204-f004:**
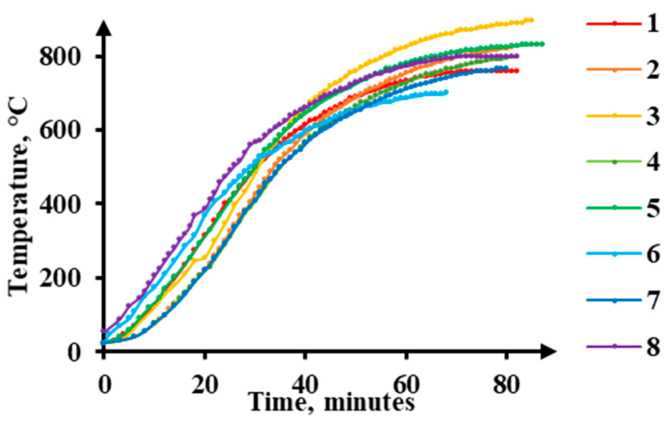
Cu-Ti-C-B composite heating schedules (the digits denote the numbers of the schedules).

**Figure 5 materials-16-03204-f005:**
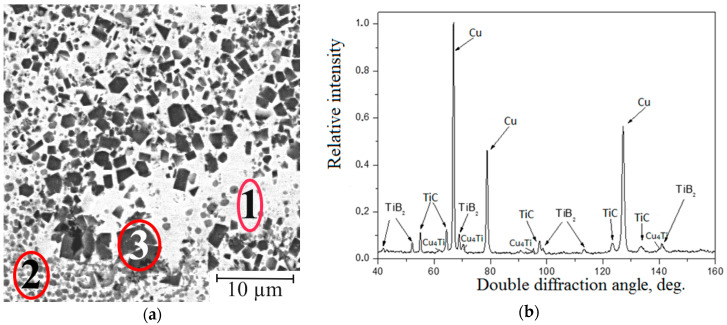
(**a**) The microstructure of the Cu-Ti-C-B composites with the following structural constituents: 1—Cu + TiC, 2—Cu + TiC + TiB_2_, 3—Cu + TiB_2_; (**b**) the X-ray pattern.

**Figure 6 materials-16-03204-f006:**
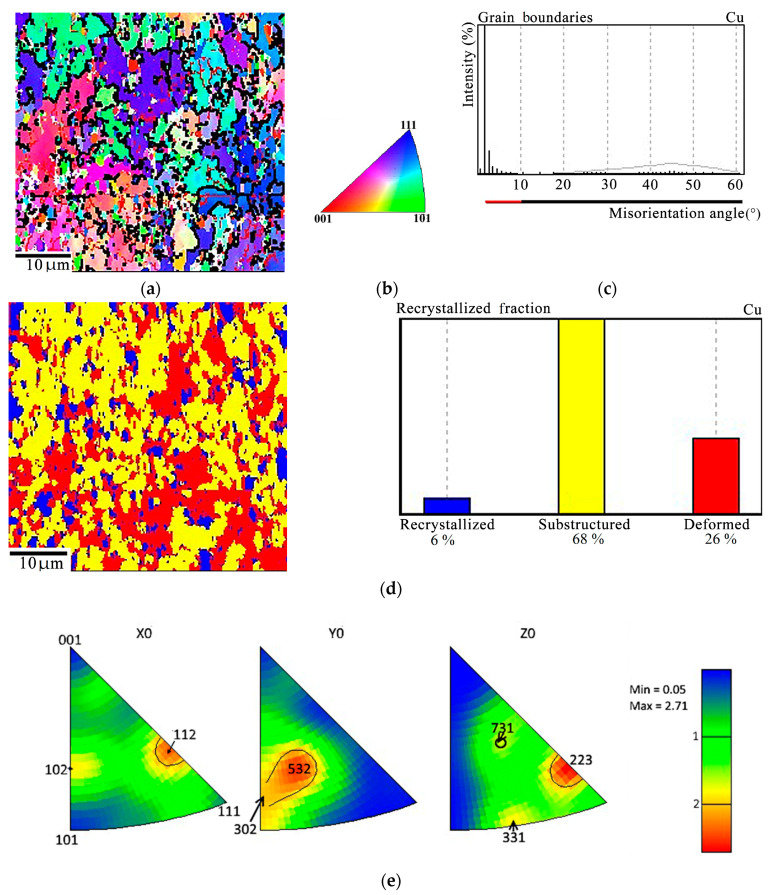
EBSD analysis of the Cu-Ti-C-B composite: (**a**) Misorientation maps with low-angle (red) and high-angle (black) boundaries in the copper matrix; (**b**) legend; (**c**) distribution of copper grains over the misorientation angles; (**d**) recrystallization map with quantitative indicators; (**e**) inverse pole figures.

**Figure 7 materials-16-03204-f007:**
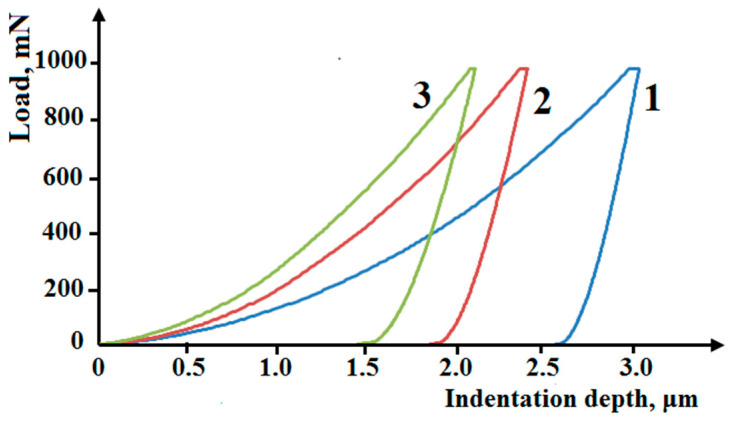
The loading diagrams of the structural constituents of the Cu-Ti-C-B. 1—Cu+TiC; 2—Cu+TiC+TiB_2_; 3—Cu+TiB_2_.

**Figure 8 materials-16-03204-f008:**
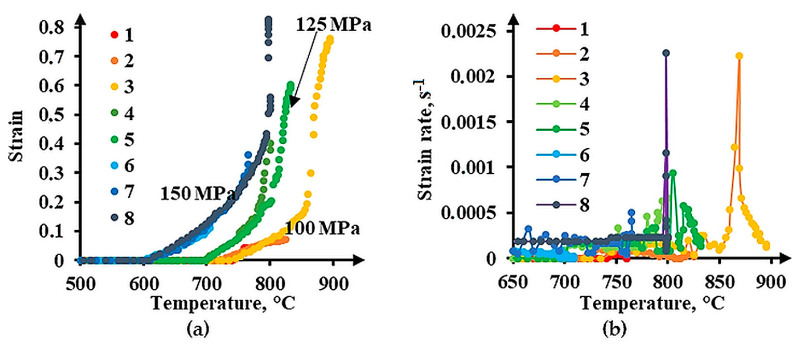
The behavior of (**a**) true strain and (**b**) strain rate in the Cu-Ti-C-B composite under heating (the digits denote the numbers of schedules in Figure 4).

**Figure 9 materials-16-03204-f009:**
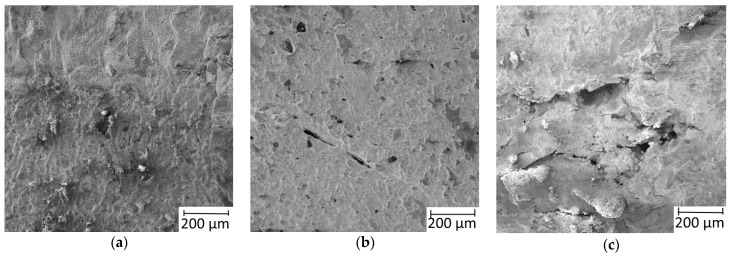
The state of the specimen surface after uniaxial deformation: (**a**) Without cracking; (**b**) with single microcracks after deformation under a load of 150 MPa at 765 °C; (**c**) with deep main cracks after deformation under a load of 100 MPa at 900 °C.

**Figure 10 materials-16-03204-f010:**
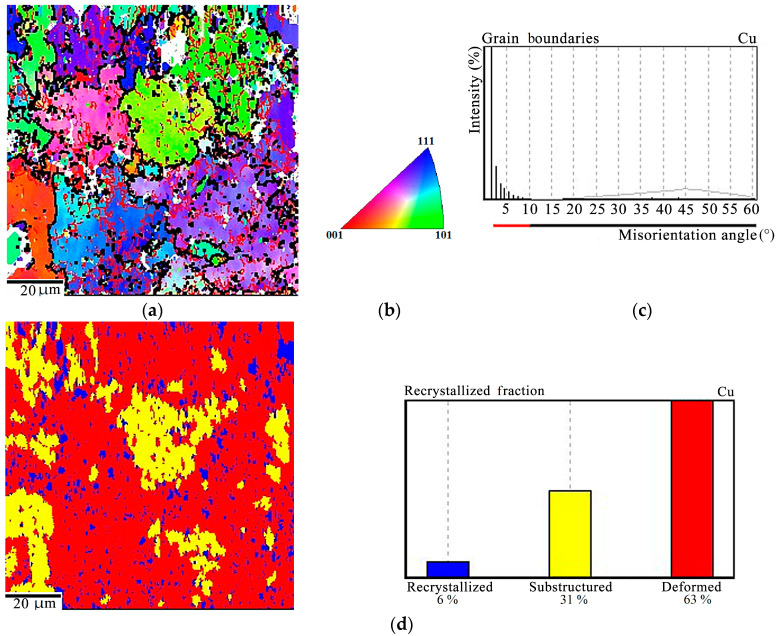
EBSD analysis of the Cu-Ti-C-B composite after deformation under a pressure of 150 MPa at 700 °C, e = 0.13: (**a**) The misorientation map with low-angle (red) and high-angle (black) boundaries in the copper matrix; (**b**) legend; (**c**) distribution of copper grains over the misorientation angles; (**d**) recrystallization map with quantitative indicators.

**Figure 11 materials-16-03204-f011:**
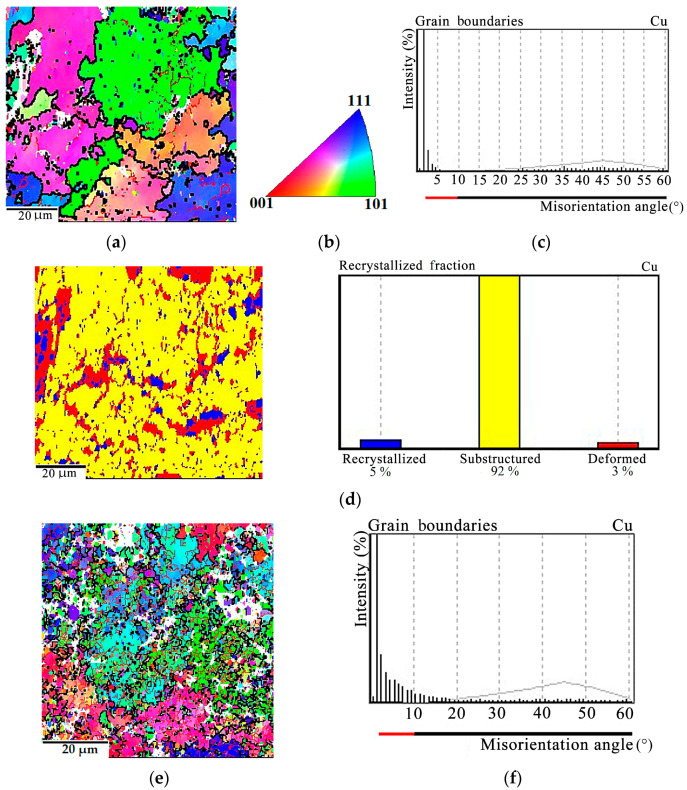
EBSD analysis of the Cu-Ti-C-B composite after deformation at 765 °C under pressures of: (**a**–**e**) 100 MPa; (**f**,**g**) 150 MPa. (**a**,**e**) Misorientation maps with low-angle boundaries shown by red and high-angle ones shown by black; (**b**) legend; (**c**,**f**) distribution of copper matrix grains over the misorientation angles; (**d**,**g**)recrystallization maps with quantitative indicators.

**Figure 12 materials-16-03204-f012:**
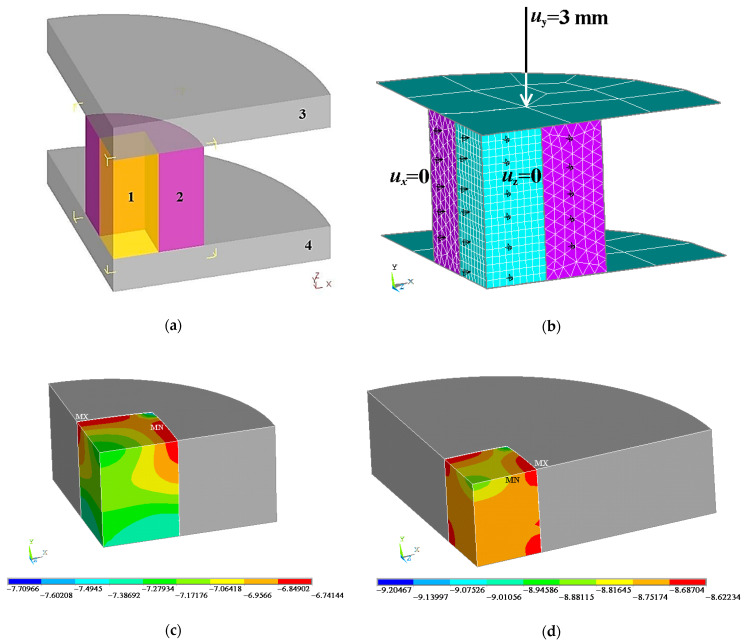
(**a**) The scheme of testing the specimen in a shell; (**b**) the finite element model of compression; (**c**) the distribution of the values of *k* in the composite with a shell diameter of 10 mm; (**d**) the distribution of the values of *k* in the composite with a shell diameter of 20 mm. 1—specimen, 2—shell, 3 and 4—plungers.

**Figure 13 materials-16-03204-f013:**
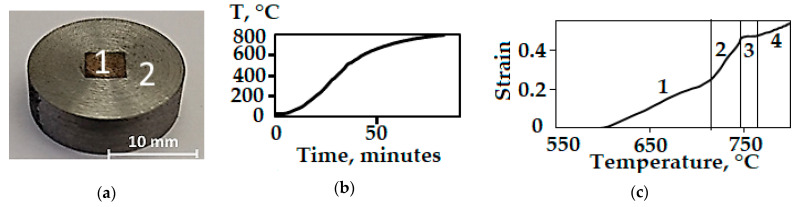
(**a**) The exterior of the composite specimen (1) in a steel shell (2); (**b**) the heating schedule; (**c**) the temperature dependence of strain.

**Figure 14 materials-16-03204-f014:**
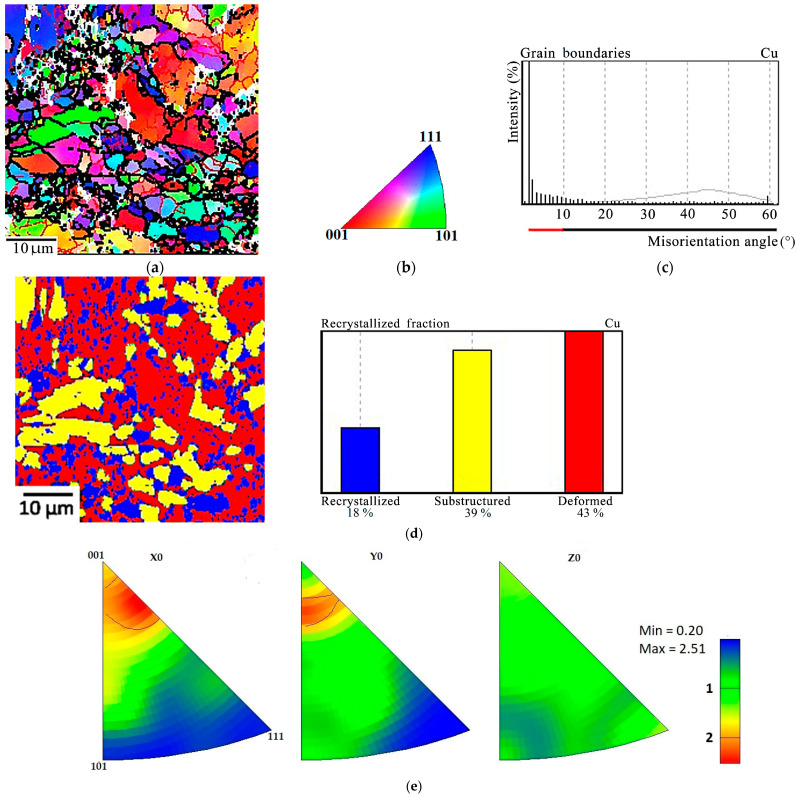
The EBSD-analysis of the Cu-Ti-C-B composite after deformation in a steel shell at 800 °C, e = 0.5: (**a**) Misorientation map with low-angle (red) and high-angle (black) boundaries in the copper matrix; (**b**) legend; (**c**) copper grain distribution over the misorientation angles; (**d**) recrystallization map with quantitative indicators; (**e**) inverse pole figures.

**Table 1 materials-16-03204-t001:** Properties of the constituents of the composite [35,36].

Components	Lattice Type	Space Group	Lattice Parameters, Å			Density, g/sm^3^	Specific Electrical Resistivity, 10^6^ Ohm∙m	Thermal Conductivity, W/m∙°	HV
			a	b	c				
Cu	FCC	Fm3m	3.62	3.62	3.62	8.96	1.72	399	35–37
TiC	FCC	Fm3m	4.34	4.34	4.34	4.93	2.0	6.80	12–25
TiB_2_	40	P6/mmm	3.03	3.03	3.23	4.52	0.09	66.4	33–34

**Table 2 materials-16-03204-t002:** The chemical compositions of the structural constituents of the Cu-Ti-C-B composite according to the X-ray microanalysis, at.%.

Constituents (Figure 5a)	Cu	Ti	С	B
1	73	14	11	2
2	49	24	20	7
3	40	25	6	32

**Table 3 materials-16-03204-t003:** The mean values of the micromechanical properties of the Cu-Ti-C-B composite constituents.

No. in Figure 5a	H_IT_, GPa (±1.5)	HV 0.1 (±1.4)	E*, GPa (±18)	W_t_, nJ (±7.3)	W_е_, nJ (±1.2)	h_max_, μm (±0.2)	R_e_, %	H_IT_/E*	φ, %	C_IT_, %
1	4.8	450	243	100	16.5	3.1	16	0.02	84	14.8
2	8.1	760	278	80.8	19.1	2.5	24	0.03	77	13.6
3	11.0	1050	293	74.5	21.9	2.2	32	0.04	71	4.7

**Table 4 materials-16-03204-t004:** Deformation conditions and obtained values of strain (**e**) and strain rate ($\xi^{*}$) (the numbers of the conditions correspond to the numbers in Figure 4).

No.	T, °C	P, MPa	e	$\xi^{*},\;s^{-1}$
1	765	100	0.04	7.9 × 10^−6^
2	825	100	0.07	1.5 × 10^−5^
3	900	100	0.76	1.5 × 10^−4^
4	800	125	0.40	8.2 × 10^−5^
5	825	125	0.60	1.2 × 10^−4^
6	700	150	0.10	3.0 × 10^−5^
7	765	150	0.36	7.5 × 10^−5^
8	800	150	0.83	1.7 × 10^−4^

**Table 5 materials-16-03204-t005:** The values of the TLEC for the Cu-Ti-C-B composite and the steel shell, α∙10^6^, K^–1^.

Т, °С	100	200	300	400	500	600	700	800	900
Composite	3.44	5.94	8.7	11.46	14.46	17.5	21.37	26.81	31.81
Steel [52]	11.9	12.8	13.5	14.1	14.6	14.9	15.2	12.5	13.5

**Table 6 materials-16-03204-t006:** The effect of the diameter of the steel shell on the value of the stress coefficient (*k*) for the Cu-Ti-C-B composite.

Diameter, mm	4	5	6	7	8	9	10	12	15
*K* _av_	−4.08	−5.83	−7.14	−7.83	−8.24	−8.51	−8.72	−8.99	−9.23
*k*_max_ − *k*_min_	2.57	2.91	0.96	0.60	0.59	0.58	0.58	0.58	0.57

## Data Availability

Not applicable.

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
