# Peer review of "Studying the Plastic Deformation of Cu-Ti-C-B Composites in a Favorable Stress State"

_materials, 2023, doi:10.3390/ma16083204_

Round 1
Reviewer 1 Report
Thank you to the authors. the work looks complete and well done.
Author Response
We welcome you!
Thank you for the higt appreciation of our work.
Best regards.
Reviewer 2 Report
I'm of the opinion that the topic of the article is very interesting. The experimental procedure was done well, but it could be said that some parts should be improved, especially from a technical point of view on the presentation:
1. Line 68: The SEM images of carbon and B4C should be provided, or a reference should be cited, at least because they are heat-sensitive constituents for the SHS process.
2. Line 156: The contrast of Figure 5(a) needs to be improved, and higher magnification images are needed for structural constituents (2 and 3).
3. The x-ray diffraction used Cr-ka as a source (not Cu ka radiation), which should be mentioned in the manuscript, and the JCPDS (or PDF card number) of standard materials (TiC, TiB2, Cu4Ti) have to be listed at the same time.
4. Figure 5(b): The "angl" should be "angle".
5. Line 342: The boundary conditions, etc., need to be specifically quantified, and the formulas of the finite element model needs to be referenced.
Author Response
Dear reviewer! Thank you for your valuable comments. We have taken into account all your comments. Corresponding corrections have been made in the text. Figures have been added or replaced. All text changes are highlighted in yellow.
Response for you comment 1:
The SEM images of carbon and boron carbideare presented on Figure 1(c,d)? corresponding text added on 68, 72.
Response for comment 2: Figure 5(a) has deen replased wiht a more contrasting one.
Response for comment 3: Information about x-ray analysis added on 83 - 84 (highlighted yellow).
Response for comment 4: The errow on Figure 5b corrected.
Response for comment 5: The element model was built using The ANSYS finit element modeling package. All formulas can be found in the manual. The link to the ANSYS packege is given in the text on 334, 335. The boundary conditions were set in displacements - 340 - 345.

Reviewer 3 Report
The authors presented an article about “Studying the Plastic Deformation of Cu-Ti-C-B composites in a Favorable Stress State.” Paper aims to improve the plastic deformation properties of copper matrix composite materials. The authors have carried out a study on an interesting topic. Today, the interest in copper matrix composite materials is quite high. In this respect, I think the article will be of interest to the reader. But I think that some points in the paper are not good.I think the paper is well organized and appropriate for the “Materials” journal, but the paper will be ready for publication after major revision.
The abstract looks good. Please include all significant numerical results.
What is the problem? Why was the manuscript written? Please explain the reason in the introduction part. In the last paragraph of the introduction, the novelty of the study and the differences from the past in detail should be expressed.
Expand the information on the production of composite materials, even if it is cited in In line 64-65. Also, prepare a visual about the production.
Specify the chemical and physical properties of the matrix and reinforcement particles used in the study in the material method section.
Give more information about Figure 2.
No information is given about the deformation test standards. Please also specify the standard information given for hardness for the deformation test.
Please give detailed information about the effects of compounds detected after x-ray analysis on composites.
Microhardness tests are generally not recommended due to the inhomogeneity of metal matrix composite materials. Why did the authors perform microhardness experiments? Please discuss in the article.
Please fix the typographical and eventual language problems in the paper.
*** Authors must consider them properly before submitting the revised manuscript. A point-by-point reply is required when the revised files are submitted.
Author Response
Dear reviewer, thank you for comments, which helped improve our article.
Our responses:
Comment 1: The abstract looks good. Please include all significant numerical resits.
Response: Changes made in lines 22 - 24.
Comment 2: What is the problem? Why was the manuscript written? Please explane the reason in the introduction part. In the last paragrapf of the intriduction, the novelty of the study and the differences from the past in detail shod be expressed
Response: The composite Cu-Ti-C-B obtained by SHS method are fundamentally new. Non one except us recived it and did not study it. In the introduction it is noted that no studies of plastic deformability of any SHS copmposites have been carry out. The aim of this manuscript is indicated in the last paragraph of the introduction. It also highlights the novelty of the study. Such research has never been done before. We added text in the lines 59 - 62.
Comment 3: Expand the unformation the production of composite materials? even if it cited in line 64-65. Also, prepare a visual about the production.
Response: We have inserted the text with information about the production of the composite - lines 85 - 92.
Comment 4: Spesify the chamical and physical properties of matrix and reinforcement particles used in the study in material method section.
Responce: Spesify the chamical and physical properties of matrix and reinforcement particles can be found in the handbooks. We belive that the properties of structural components, which are discussed in detail in the manuscript, are more important. To solve the goal set in the manuscript the micromechanical properties that we studed are very imtortent. We are currently preparing a manuscript about the phyzical properties of the composite Cu-Ti-C-B, where it would be more appropriate to provide data on the properties of matrix and reinforcing particles.
Comment 5: Give more information about Figure 2.
Responce: The text already contans all the information adout Figure 2.
Comment 6: No information is given about the deformation test standard. Please also spesify the standard information given for hardness for the defobmation test.
Responce: Deformation tests are not standard. The original techniques are described in the lines 135 - 162 and 335-344. The standard for hardness measuremant is given in the line 102.
Comment 7: Please give detailed informstion about the effects of compounds detected after x-ray analysis on composites.
Response: The copper matrix provides a bond between the reinforcing particles. Reinforcing particles strengthen the composite.
Comment 8: Microhardness tests are generally not recommended due to the infomogeneity of metal matrix composite materials. Why did the authors perform microhardness experiments? Please discuss in the article.
Response: We did not limit oueselves to the determination of microhardness. Determination of the micromechanical properties of the structural components is necessary to assess the possibility of plastic deformation of composite. The authors of many manuscripts use the micromechanical properties of heterophase composites. We added text in the line 131-134.
Comment 9: Please fix typographical and eventual lsnguage problems in the paper.
Response: We have corrected spelling errors and tried to improve the style of the text. All fixes are higlighted in blue.

Round 2
Reviewer 3 Report
The authors did not undertake any study with the corrections given below. Therefore, the paper quality is currently not suitable for Materials.
· Authors are required to prepare a figure that includes the material production stage.
· The chemical and physical properties of the matrix and reinforcement particles must be specified in the paper.
· The positive and negative properties of the compounds obtained after the XRD test results on the material should be discussed.
These corrections should be made and stated in the paper. Otherwise, I would like to say that it would not be appropriate to publish the paper.
Author Response
Dear reviewer, we have added new information ti the text.
Comment 1: Authors are required to prepare a figure that includes the material production stage.
Response: Figure 2a inserted into the text, lines 122, 123, 127.
Comment 2: The chamical and physical properties of the matrix and reinforcment particles must be spesified in the paper.
Responce: The properties added to the text - lines 94 - 98, Table 1.
Comment 3: The positive and negative properties of the compounds obtained after the XRD test results on the material should be discussed.
Response: The positive and negative properties of the compounds are discussed in the lines 200-213.
